# Mechanisms of Primary and Secondary Resistance to Immune Checkpoint Inhibitors in Cancer

**DOI:** 10.3390/medsci7020014

**Published:** 2019-01-22

**Authors:** Tiffany Seto, Danny Sam, Minggui Pan

**Affiliations:** 1Internal Medicine Residency Program; tiffany.seto@kp.org (T.S.); danny.sam@kp.org (D.S.); 2Department of Oncology and Hematology, Kaiser Permanente, Santa Clara, CA 95051, USA; 3Kaiser Permanente Division of Research, Oakland, CA 94612, USA

**Keywords:** Immune checkpoint inhibitor, Primary resistance, Secondary resistance, Cancer immunotherapy, PD1, PD-L1, T cells, TCR, CTLA4.

## Abstract

Immune checkpoint inhibitors (ICPis) have revolutionized cancer therapy with broad activities against a wide range of malignancies. However, in many malignancies their efficacy remains limited due to the primary resistance. Furthermore, a high percentage of patients develop progression due to the secondary resistance even after obtaining a response or achieving a stable disease. In this review, we will discuss the mechanisms that underlie the primary and secondary resistance to ICPis in cancer immunotherapy and provide an overview to impart a broad understanding of the critical issues that are encountered in clinical oncology practice.

## 1. Introduction

Immune checkpoint inhibitors (ICPis) have demonstrated a response rate ranging from approximately 80% for refractory Hodgkin lymphoma to less than 20% for most of the other malignancies. For several types of cancer, the response rate has been in the single-digit range or only occasionally observed. For example, few responses have been observed with respect to breast cancer (especially in hormone receptor-positive breast cancer), prostate cancer, pancreatic cancer, colorectal cancer (except for those with mismatched DNA repair protein deficiency), and most of the soft tissue and bone sarcomas.

Numerous factors impact the efficacy of the ICPis in cancer treatment [1,2]. These factors can be categorically divided into either the host or the tumor factors. The host factors include a patient’s gut microbiome, physical performance status, co-existing morbidities, immune state, medication use, and so on. The tumor factors include the tumor’s underlying biology such as histology and aneuploidy, tumor microenvironment, type of gene mutations harbored, and the burden of tumor mutation, among others. Here, we provide a review of the factors that are related to primary or secondary resistance to ICPis and discuss their potential mechanisms. Understanding these factors may provide important insight for guiding clinical practice and further research on cancer immunotherapy.

## 2. Host Factors

In Table 1 we have summarized the host factors and the tumor factors that potentially impact the efficacy of ICPis. For the host factors, the gut microbiota has been found to significantly impact the response to the ICPis in both mice and humans [3,4,5]. Melanoma-bearing mice treated with antibiotics or fed with germ-free diets failed to respond to the anti-CTLA blockade, but their response was restored by the gavage of *Bacteroides fragilis* [6]. Another study showed a correlation between the clinical responses to ICPis and the relative abundance of *Akkermansia muciniphila*, and the fecal transplantation of this bacterium restored the response of mice to the anti-PD-1 blockade [5]. In patients with metastatic melanoma treated with an anti-PD1 antibody, the relative abundance of the *Ruminococcaceae* family was associated with a better response rate (RR) and longer progression-free survival (PFS) [3]. The influence of gut microbiota on the response of the host’s malignancy to ICPis is consistent with the regulatory roles of the gut flora on the host’s immune reactive state [4,7]. The gut microbiome plays a role not only in inflammatory bowel disease, but also the diseases outside of the bowel [8]. The recognition of such a role is increasingly expanding including the immune modulation of the skin and liver, as well as fat metabolism (obesity) [9,10,11,12].

There is evidence raising concerns that the use of antibiotics before or during treatment with ICPis could be associated with the compromised efficacy of such ICPis due to the altering of the gut flora. The retrospective data of 303 patients (mostly melanoma and lung and kidney cancer patients) from the Christie National Health Service (NHS) Foundation of the United Kingdom presented at the 2018 American Society of Clinical Oncology (ASCO) annual meeting showed worse PFS and overall survival (OS) for those patients treated with antibiotics (mostly β-lactam or macrolides) before or during treatment with an ICPi [13]. A French study found that patients with lung cancer or renal cell carcinoma treated with a β-lactam for pneumonia or a quinolone for urinary tract infection were associated with shorter PFS and OS [14]. The importance of gut microbiota in association with the efficacy of ICPis also raises a question about the potential impact of a patient’s dietary styles. For example, a study using animal models found that dietary protein restriction improved the tumoricidal activity of tumor-associated macrophage (TAM) [15]. What about probiotics or red meat consumption during the treatment with an ICPi? How would a purely vegetable diet impact the efficacy of ICPis? These questions may require large epidemiological studies to shed light on these issues.

The impact of steroid use on the efficacy of ICPis continues to be debated. A retrospective review of more than 600 patients with metastatic non-small cell lung cancer, treated at memorial Sloan Kettering Cancer Center and Gustave Roussey Cancer Center, showed that the baseline use of 10 mg or more of prednisone or its equivalent daily at the beginning of ICPi treatment was associated with worse PFS and OS [16]. The meta-analysis by Garant et al. found that the concomitant use of a steroid with ICPis did not appear to compromise patient outcomes [17]. Of course, both of the aforementioned studies were not prospective and had their own inherent pitfalls. Mechanistically, it appears that dexamethasone diminishes T-cell function by impairing the CD28 costimulatory pathway, while the blockade of CTLA-4, but not PD-1, can partially prevent this impairment [18]. Though still debatable, it would be wise to minimize the use of steroids when possible. The similar clinical situations are frequently encountered in patients who require steroid use for preventing angiogenic edema because of radiation to the brain for metastatic disease. In those patients, following a tapering schedule relevant to the prevention of brain edema is appropriate. For patients who require steroid use as part of a chemotherapy pre-medication regimen, a shorter duration and lower dose may be considered when possible without risking increased chemotherapy toxicities.

Physical performance status (PS) and comorbid conditions are well-established factors associated with responses to chemotherapy. It would not be surprising to see a similar correlation with ICPis, as these two host factors can be closely correlated with a patient’s immune state. Physiologically poor PS and serious comorbidities are likely to be associated with an immune suppressive state that impedes the activation of the host’s immune response by such ICPis. However, in a meta-analysis, PS was not found to be associated with OS in patients treated with ICPis [19]. In a phase-II trial with anti-PD-L1 antibody atezolizumab in metastatic urothelial carcinoma, patients with PSs of 2 were found to obtain similar RRs compared with all the patients in the trial [20]. In our study of patients with metastatic melanoma treated with nivolumab or pembrolizumab, we observed that poor PS and comorbidities were associated with a more rapid progression of disease [21]. However, these results were retrospective and likely indicate that the effects of PS and comorbid conditions on cancer immunotherapy are more complex than that have been observed in systemic chemotherapy and are possibly compounded by the other clinical factors. A recent meta-analysis looked at the association of sex and the efficacy of ICPis and found no difference in all the subsets analyzed [22].

The association of certain host biomarkers with the relative benefits to ICPis has also been investigated [23]. These biomarkers include serum lactic acid dehydrogenase (LDH), C-reactive protein (CRP), eosinophil count, monocyte count, neutrophil count, and platelet count, as well as lymphocyte count. Diem et al. showed that elevated LDH at the baseline was associated with worse survival and poorer response to anti-PD1 therapy, while the reduction of LDH levels during treatment with anti-PD1 was associated with early response [24]. Our study of patients with metastatic melanoma showed that a high neutrophil count and a high platelet count were associated with more rapid disease progression, while a high serum lymphocyte count was associated with slower disease progression [21]. Patients with a baseline neutrophil count of >5500/µL had twice the risk of progression during the treatment with nivolumab or pembrolizumab compared with patients whose baseline neutrophil count was <3900/µL, regardless of BRAF mutation status. A platelet count of >304,000/µL was similarly associated with twice the risk of progression compared with a platelet count of <215,000/µL, while a lymphocyte count of >1716/µL was associated with half of the risk of progression compared with a lymphocyte count of <1120/µL. We have also observed a similar association in patients with metastatic non-small cell lung cancer (unpublished data). Elevated levels of inflammatory biomarkers may be associated with increased levels of regulatory T cells (Tregs) in peripheral blood and in the tumor microenvironment that can lead to a heightened immune suppressive state of the host. The simple peripheral blood counts may be valuable in guiding day-to-day practice when using ICPis. For patients with a high neutrophil count and/or a high platelet count and/or a low lymphocyte count, we recommended more frequent imaging studies to capture the progression earlier rather than later. Our data are consistent with a recent meta-analysis that included studies of patients with metastatic melanoma, non-small cell lung cancer, and urothelial carcinoma and showed that a high neutrophil-to-lymphocyte ratio (NLR) was associated with twice the risk of mortality and progression during treatment with ICPis [25]. Another study showed that the decline of NLR 6 weeks after the initiation of an ICPi treatment was associated with better outcomes in patients with metastatic renal cell carcinoma [26], suggesting that the improvement of such a biomarker during treatment can be an early and valid predictor of benefit, which is consistent with the data from the study by Diem et al. [24,26]. A French study looked at both NLR (>3.0) and LDH (>upper-normal range) prior to treatment with an ICPi and found that the elevated index modeled from these two biomarkers (LIPI) was associated with shorter PFS [27]. Thus, a combination of peripheral blood counts and inflammatory markers may provide valuable guidance for clinical practice using ICPis. However, the value of biomarkers in association with the combination of ICPi with chemotherapy remains to be investigated.

## 3. Tumor Factors

The role of the tumor factors in the primary and secondary resistance to ICPis is extremely complex and largely determined by the underlying tumor biology and tumor microenvironment (Table 1). The tumor microenvironment is modulated by numerous factors including the expression of anergic and immunosuppressive proteins such as PD-L1, indoleamine-2,3-dioxygenase (IDO), immunosuppressive cytokines, and the infiltration of myeloid-derived suppressive cells (MDSCs), including macrophages and dendritic cells, as well as tumor-infiltrating T lymphocytes (TILs), neoantigen load, and mutation burden which positively regulate the immune response [28,29,30,31]. In addition to the histology, tumor aneuploidy has been associated with intrinsic resistance to ICPis due to immune evasion, as patients with metastatic melanoma with highly aneuploid tumors were less likely to respond to ICPis [32]. These tumors with abnormal numbers of chromosomes or chromosome segments expressed very few markers for immune cell infiltration [32]. This discovery opens up a new front for studying biomarkers and immune response. Exactly how much the specific histology of a tumor contributes to the response or lack of response to ICPis remains a question, as one may argue that the tumor microenvironment determines sensitivity to ICPis rather than the histology itself. This notion is supported by the high response rate observed across all the malignancies harboring mismatched DNA repair protein deficiency [33,34,35].

The most commonly used biomarker for predicting response to ICPis is the expression of PD-L1 on the tumor cells. While tumors with zero expression of PD-L1 can occasionally respond to ICPis, high expression of PD-L1 has generally been associated with higher RR and better PFS [36,37]. The blockade of PD-L1 removes the brake on host T lymphocytes and enables them to recognize tumor cells for cellular destruction [38]. PD-L1 expression is often associated with increased TILs [39,40,41]. Some retrospective studies have revealed that PD-L1 expression and/or TILs are associated with worse outcomes, while others have shown the opposite, in the absence of treatment with ICPis [41,42,43]. Neoantigen load is often associated with PD-L1 expression, as well as the presence of TILs, which are critical for establishing a response to ICPis [44]. There is a broad variation of PD-L1 expression across different histology, from more than 50% in some malignancies to less than 20% in many others. An analysis of the results from the Keynote-028 study revealed that the tumors with high PD-L1 expression, high expression of T-cell-inflamed genes, and high tumor mutation burden were associated with high benefit from pembrolizumab across several different tumor types [45]. Facilitating the infiltration of tumors by TILs has been an ongoing strategy for enhancing the response to ICPis [46,47]. The phase-I/II trials with NKTR-214, a pegylated interleukin-2 (IL-2) showing an enhanced response rate on top of nivolumab in several solid tumor types with expanded TILs, have attracted much attention and many doubts [48], while the phase-III trial is ongoing with metastatic melanoma.

The mechanisms of T-cell activation by ICPis remain incompletely understood. Some studies have shown that PD1 inhibits CD28 co-stimulation of T cells and that the activation of T cells by ICPis is CD28-dependent. The activation of PD1 on T cells by PD-L1 recruits shp2 tyrosine phosphatase to dephosphorylate CD28, leading to its inactivation, while the deletion of CD28 causes the failure of the T-cell activation by the ICPis [49,50]. This is not surprising as the previous study showed that co-stimulation of T cells through CD28 used the same signaling pathway by the T cell receptor (TCR) and the role of CD28 co-stimulation was to maintain nuclear occupancy of nuclear factors of activated T cells (NFATs) [51]. The NFAT proteins are critical for mediating and maintaining TCR signaling, as well as the transcription of a whole battery of cytokines upon T-cell activation [52,53].

IDO was initially viewed as a promising target and biomarker, as high levels of IDO expression were associated with shorter PFS and poorer outcomes [54]. IDO induces inflammation within the tumor microenvironment, depletes the tryptophan required by cytotoxic T-cells, and induces the conversion of naïve T-cells to Tregs, thereby promoting a tolerogenic and immunosuppressive state. Unfortunately, recent clinical trials with IDO inhibitors have failed to show added efficacy on top of ICPis, raising the needs to reevaluate its mechanisms and clinical trial design.

The production of TGF-β by the fibroblasts in the tumor microenvironment impairs the host response to ICPis—another key tumor factor that presents opportunities for therapeutic targeting [55]. Other immunosuppressive cytokines play key roles as well in blocking the response to ICPis [56,57]. A preclinical study showed that blocking both IL-6 and TGF-β enhanced the activity of ICPis [58]. Another study showed that targeting IL-6 increased the expression of PD-L1 on the melanoma cells and attracted the expression of T-cell inflammatory cytokines in the tumor microenvironment and enhanced the activity of anti-PD-L1 therapy [59]. An animal study showed that IL-10 release following ICPi therapy may be a resistance mechanism for ovarian cancer to evade immune detection [60]. In contrast, other studies have shown that a pegylated IL-10 (AM0010) stimulates the proliferation of cytotoxic T-cells and enhances the efficacy of ICPis in early clinical trials [61,62,63]. Clinical trials with IL-12 have also shown interesting results [64]. An early clinical trial has shown promising activity with an IL-15 superagonist in combination with nivolumab [65]. Many clinical studies are exploring bi-specific fusion antibodies that target two surface molecules simultaneously. A recently published phase-I study with a bi-functional fusion antibody targeting both TGF-β and PD-L1 showed four responses and two prolonged stable disease in 19 heavily pretreated patients (cervical cancer, pancreatic cancer, etc.) [66].

Another strategy with the potential to reverse the primary resistance and enhance the activity of ICPis is to combine the agents that stimulate innate immunity, including toll-like receptor (TLR) agonists. Innate immunity is likely the main mechanism responsible for the spontaneous regression of malignancies following an episode of infection, either induced or coincidental, with Coley’s inoculation of erysipelas as the most cited since 1893 [67]. Several pre-clinical studies have shown the synergistic effect of a TLR agonist with an ICPi [68]. A number of clinical trials are testing this hypothesis, using TLR4, 7/8, and 9 agonists and others by intra-tumoral injection [69,70,71,72]. In a phase-I/II study with 15 patients with indolent lymphoma injected with a TLR9 agonist, five responses were observed [73]. A recent phase-Ib trial showed evidence that a TLR9 agonist might be able to reverse the resistance to pembrolizumab in patients with metastatic melanoma [74]. The inhibitors and stimulators of several other co-inhibitory and co-stimulatory molecules (STING, 4-1BB, OX40, etc.) are being intensely tested as well [75,76,77].

High tumor mutation burden (TMB) is associated with a higher RR in many retrospective studies [78]. High TMB leads to the increased release of neoantigens in the tumor microenvironment, thereby increasing the antigenic presentation and attracting host T cells to infiltrate the tumors [79]. The high TMB can also be reflective of increased genomic instability; this is consistent with the high RR seen in patients with mismatched DNA repair deficiency regardless of tumor histology [33,34,35,44]. A recent study found that alterations of the genes involved in the DNA damage repair and response, including *ATM*, *BRCA2*, *ERCC2*, *FANCA*, *MSH6*, *POLE*, and so on, were associated with a significantly higher RR to ICPis in metastatic bladder cancer [80]. A recently published phase-III clinical trial showed a significant extension of PFS for patients with metastatic non-small cell lung cancer with high TMB treated with ipilimumab plus nivolumab compared to those treated with chemotherapy [81]. Tumors that display a hypermutable phenotype due to the mutation of *POLE* are also associated with higher response rates to ICPis [82,83]. The exploitation of neoantigens for immunotherapy, including developing neoantigen vaccines, is being intensely investigated in melanoma, glioblastoma, and other types of cancer [84,85]. In contrast to the solid tumors in which high TMB is associated with a response to ICPis, high TMB and elevated neoantigen load were shown to be associated with poor survival in patients with multiple myeloma [86]. Taking advantage of neoantigens or specific cell surface proteins produced by the tumors to design chimeric antigen T-cell receptor therapy (CAR-T) presents an exciting future [87,88,89,90,91,92]. Because one of the mechanisms of CAR-T failure is T-cell exhaustion due to the increased expression of PD1 on T-cells, the attempt is being made to test if the addition of ICPi would prevent or reverse this failure [93]. This combination will likely require sophisticated timing of combining an ICPi during the course of CAR-T therapy to avoid excessive toxicities and to achieve maximum tumoricidal synergy.

## 4. Mechanisms of Primary Resistance

While many host factors are associated with the primary resistance to ICPis, the most critical ones are tumor biology and tumor microenvironment. Host factors in some unique patient populations, including patients with autoimmune disease, histories of organ or bone marrow transplantation, impaired organ function, extremely young or old age, pregnancy, or chronic viral, bacterial or fungal infections, may have a significant impact on the efficacy and toxicities of ICPis [94,95]. Many of these factors are also part of patient comorbidities. For patients with organ transplants, some case reports have shown safe administration of an ICPi with efficacy, while others have shown rapid deterioration of kidney functions due to rejection by patients with histories of kidney transplants [95,96,97,98]. For patients with autoimmune disease, the flare of autoimmune disease has been observed in at least one-third of the patients treated with an ICPi, while reasonable response rates have also been observed (approximately 20% in patients with melanoma and lung cancer) [99,100,101]. In patients with metastatic non-small lung cancer with preexisting autoimmune disease, ICPis have been shown to be similarly effective with similar survival rates [102,103]. Little is known about the impact of the other comorbidities on the efficacy of ICPis; for example, patients with diabetes are known to have impaired immunity. Is there an association between the level of control of blood glucose (level of hemoglobin A1c) and the efficacy of ICPis?

Overcoming the primary resistance to ICPis has been an area of intense investigation in cancer immunotherapy. Combination approaches with the other modalities and agents have shown certain levels of success. Using radiotherapy to produce an abscopal effect has been well described and explored [104,105,106]. Golden et al. showed a 27% abscopal effect using radiotherapy and granulocyte-macrophage colony stimulating factor (GM-CSF) in solid tumors that had failed many lines of therapy [107]. Ribas et al. demonstrated an enhanced response rate when pembrolizumab was combined with the talimogene laherparepvec treatment in patients with metastatic melanoma in a phase-Ib trial [108]. Talimogene laherparepvec is an oncolytic virotherapy approved by the U.S. FDA for local injection for locally advanced melanoma [109]. The combination of low-dose ipilimumab with nivolumab has shown added activities in melanoma, renal cell carcinoma, and non-small cell lung cancer with high TMB with improved PFS and OS [81,110,111]. The combination of chemotherapy with pembrolizumab has shown greater improved survival than chemotherapy alone in patients with metastatic non-small cell lung cancer in a first-line setting [112]. Numerous clinical trials are testing combination approaches with chemotherapy (JAVELIN trials, etc.), targeted therapeutics, adoptive cell therapy, cancer vaccines, and so on with some very exciting results.

## 5. Mechanisms of Secondary Resistance

The mechanisms of the secondary resistance to ICPis are primarily an evolutionary consequence of malignancies under the pressure of an immune attack. Many malignancies develop upregulated PD-L1 expression during the course of treatment to overcome the blockade by ICPis or produce immunosuppressive cytokines and/or T-cell inhibitory receptors to induce T-cell exhaustion, while some tumors develop mutations of certain critical genes that regulate the response to ICPis [113,114,115,116,117,118]. 

Several gene mutations have been found to confer secondary resistance to ICPis. These mutations primarily involve the immune effector signaling pathways, such as the interferon response pathway and antigen presentation proteins. Gao et al. found that certain melanoma patients who failed to respond to ipilimumab were associated with cellular defect of genes in the γ-interferon pathway and demonstrated that knockdown of γ-interferon pathway genes in animals impaired their responses to anti-CTLA4 antibody therapy [119]. Zaretsky et al. identified the deletion of *JAK1* and *JAK2* genes in two melanoma patients, respectively, who progressed on pembrolizumab after a response and a truncating mutation in the gene encoding the antigen-presenting protein β-2-microglobulin (*B2M*) in a third patient [120]. The loss of function of *JAK1* and *JAK2* leads to impaired interferon signaling, which is required for the effective activation of the cytotoxic T-cells stimulated by the checkpoint inhibition. The loss of *B2M* leads to defective antigen presentation. A case report identified the *PTEN* gene mutation as the cause of progression in a patient with uterine leiomyosarcoma who initially responded to pembrolizumab. The loss of *PTEN* led to the loss of two neoantigens thought to be responsible for the response to ICPis [120]. These discoveries are extremely exciting and provide the kind of critical information necessary to understand the mechanisms of the secondary resistance to ICPis and hence establish a platform for the further development of enhanced immunotherapy. The strategies for overcoming secondary resistance are evolving, including the approaches for enhancing the persistence of memory T-cells, preventing T-cell exhaustion, etc. [121,122]. For patients whose tumors develop the mutation of the genes governing the interferon signaling pathway, it would be important to determine if a response can still be obtained by combining a TLR agonist or other stimulators of the innate immunity. The past reports of the cure or long-term remission of a cancer induced by bacterial infection are primarily a function of inducing innate immunity. Would such an induction of innate immunity still be effective for patients whose malignancy has progressed through the treatment of ICPis? 

## 6. Concluding Remarks

Cancer immunotherapy has transformed the natural history of cancer, making the cure possible for many patients with advanced diseases that were, in the past, not curable. However, cancer immunotherapy is still in its early stages before achieving its full potential. Many setbacks, such as the recent failure with the IDO inhibitors, will continue to litter the path, but the future is incredibly hopeful. The agonists for IL-15, IL-2, and TLRs have shown very interesting but mixed results. Targeting other immune checkpoint regulators is generating promising data as well. Elucidating the mechanisms for regulating the response to ICPis from both the perspective of the host and the tumor will continue to provide insightful information to guide cancer treatment.

## Figures and Tables

**Table 1 medsci-07-00014-t001:** Factors associated with primary and secondary resistance to immune checkpoint inhibitors (ICPis).

Host Factors	Tumor Factors
Gut microbiota	Tumor biology (aneuploidy, histology, etc.)
Performance status	Tumor microenvironment (PD-L1 expression, neoantigen load, immunosuppressive cytokines, tumor-associated microphages (TAMs), tumor-infiltrating T lymphocytes (TILs), etc.)
Immune status (inflammatory biomarkers, regulatory T-cells (Tregs), etc.)	
	Gene mutations (γ-interferon pathway, JAK1, JAK2, PTEN, B2M, etc.)
Comorbidities	
	Mutation burden (*POLE* mutation, DNA mismatched repair protein deficiency, etc.)
Use of antibiotics	
Use of steroids	

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
