# Peer review of "Mechanisms of Primary and Secondary Resistance to Immune Checkpoint Inhibitors in Cancer"

_medsci, 2019, doi:10.3390/medsci7020014_

Round 1
Reviewer 1 Report
This review paper by Seto et al entitled "Mechanisms of primary and secondary resistance to immune checkpoint inhibitors in cancer: a mini-review" summarized some host factors and potential tumor factors that may contribute to the resistance to the cancer therapy with immune checkpoint inhibitors. Overall the mini-review provided a simplified list on both host factors and tumor factors that may play potential roles in association to therapy resistance. The mini-review intended to help clinical oncologists in clinical trial design and patient selections for immune checkpoint inhibitors.
It has been well documented that response rate to ICIs ranged from a few percentage to 80% depended on tumor types and other factors. With the FDA approvals of several ICIs for different indications, more real world experience in ICIs will give much needed true response rates. The interactions between basic immunology research and clinical oncological research will provide new targets for combination immunotherapy.
In the host factors, the authors listed gut microbiota in a prominent position. Steroid use affected the microbiota. This should lead to greater efforts in searching for better second generation of ICIs with better anti-tumor efficacy but less immunotherapy related adverse events (irAE). Steroid use is associated with higher irAE. Some cancer types have chemotherapy or radiation therapy in conjunction or prior to ICI therapy. These therapies severely reduce the gut microbiota quantities and diversities. These factors may warrant further discussion.
In the tumor factors, the mutation burden status and tumor microenvironment may need more attention for ICI efficacy. Many agents in pre-clinical or clinical research stages are trying to convert cold tumors to hot tumors.
In general, it is not clear at current stage whether the listed factors in mini-review have direct impact for clinical practice guidance.
Author Response
Thanks for the positive comment. We have made substantial revision with expanded discussion of many more studies and references to a comprehensive review article. We have expanded with much more discussion on the host factors and tumor factors and the mechanisms including many pre-clinical studies. we have included 121 references now.
Reviewer 2 Report
This is a concise yet comprehensive minireview on pitfalls of modern cancer immunotherapy.
The schematic subdivision in host and tumor-related conditions affecting the therapeutic outcome is interesting and well pointed.
The text, however, must be thoroughly revised and improved before publication.
to mention just a few points that must be amended:
lines 10-11...is limited in most malignancies due....
lines 32-33...important to guide......something is missing!!??
lines 57-58...showed significant that high........please rephrase it!!
Author Response
Thanks for pointing out the errors. We have made the corrections and expanded the review article to a comprehensive one with many more references and in-depth discussion.
Reviewer 3 Report
Overall, this is a well written and concise mini-review of the potential mechanisms of resistance to immune checkpoint inhibitor therapy in cancer. A few things need the authors’ attention:
1. Introduction is lacking any references- please provide these.
2. Lines 48-49 state that performance status and comorbid conditions are well-established factors associated with response to ICI. However, but only one study is discussed and the data are not as robust as implied by the authors. This was a retrospective cohort study in ~100 melanoma patients treated with nivolumab. Hazard ratios by comorbidity index did not reach statistical significance. Hazard ratios by PS was significant but unclear if p-values were adjusted for multiple comparisons. In general, this study is difficult to interpret since there was no control cohort that did not receive ICI, so it is not possible to determine whether these are poor prognostic indicators in general or if they are uniquely associated with responses to ICI. Please discuss additional evidence supporting the statement or revise to reflect that these are emerging findings that warrant further investigations.
3. The authors distinguish between host versus tumor factors associated with resistance to ICI, but they do not consistently distinguish between primary versus secondary resistance mechanisms although this is the focus based on the article title.
4. Some editing is needed but it isn’t too extensive.
Author Response
We have added references in the Introduction.
2. We have expanded in-depth discussion on the PS and comorbidities substantially with references several more studies.
We have create separate sections for "primary resistance" and "Secondary resistance" with expanded in-depth discussion.
4. we have corrected the errors and grammars.